---

biomimetics/computational mechanics/computer-aided design

bioinspired gliding wing prototypes, finite-element analysis, flexural stiffness, additive manufacturing, vacuum thermoforming

**Author for correspondence:**
Hamid Isakhani
e-mail: hamid.isakhani@gmail.com, hisakhani@lincoln.ac.uk

# Towards locust-inspired gliding wing prototypes for micro aerial vehicle applications

Hamid Isakhani[1], Caihua Xiong[2], Wenbin Chen[2] and Shigang Yue[1,3]

[1]The Computational Intelligence Lab (CIL), School of Computer Science, University of Lincoln, LN6 7TS Lincoln, UK
[2]The State Key Laboratory of Digital Manufacturing Equipment and Technology, School of Mechanical Science and Engineering, Huazhong University of Science and Technology, Wuhan 430074, People's Republic of China
[3]Machine Life and Intelligence Research Centre, Guangzhou University, Guangzhou 510006, People's Republic of China

HI, 0000-0002-7443-371X; CX, 0000-0003-2326-0289

In aviation, gliding is the most economical mode of flight explicitly appreciated by natural fliers. They achieve it by high-performance wing structures evolved over millions of years in nature. Among other prehistoric beings, locust is a perfect example of such natural glider capable of endured transatlantic flights that could inspire a practical solution to achieve similar capabilities on micro aerial vehicles. An investigation in this study demonstrates the effects of haemolymph on the flexibility of several flying insect wings proving that many species exist with further simplistic yet well-designed wing structures. However, biomimicry of such aerodynamic and structural properties is hindered by the limitations of modern as well as conventional fabrication technologies in terms of availability and precision, respectively. Therefore, here we adopt finite-element analysis to investigate the manufacturing-worthiness of a three-dimensional digitally reconstructed locust wing, and propose novel combinations of economical and readily available manufacturing methods to develop the model into prototypes that are structurally similar to their counterparts in nature while maintaining the optimum gliding ratio previously obtained in the aerodynamic simulations. The former is assessed here via an experimental analysis of the flexural stiffness and maximum deformation rate as $EI_s = 1.34 \times 10^{-4} \, Nm^2$, $EI_c = 5.67 \times 10^{-6} \, Nm^2$ and greater than 148.2%, respectively. Ultimately, a comparative study of the mechanical properties reveals the feasibility of each prototype for gliding micro aerial vehicle applications.

# 1. Introduction

A well-known crucial challenge in aeronautics is to conserve energy as much as possible in long-range flights. One straightforward solution is to extend the aerial vehicle's gliding capability, which in turn facilitates acoustically (reduced propulsion noise) and aerodynamically (laminar flow) comfortable distant flights, apart from reducing emissions (carbon footprints) significantly. To achieve this, further consideration of the available alternatives such as a bioinspired solution is advantageous to the study of this flight mode. One such source of inspiration found in nature, is a flying insect commonly called a desert locust (*Schistocerca gregaria*) capable of excellent gliding but infamous for their disastrous presence in the agricultural fields. Focusing on the positive traits of this insect, we are inspired by their exceptional swarming flight endured via gliding [1,2], that is also well coordinated and collision-free [3–5].

Since the mid-twentieth century, other cross-disciplinary researchers were also inspired by the same traits to reveal the underlying physiological characteristics contributing to this insect's near-perfect flight performance. In 1951, Waloff & Rainey [6,7] presented a quantitative study on the East African desert locust's swarming behaviour concluding that their endured gliding is directly associated with their tendency to fly in swarms. A few years later, Weis-Fogh [8] commenced a detailed study on the locust's aerodynamic footprint that is continuously pursued by other researchers presented in the contemporary literature [9–14]. Recent studies suggest that the wing mechanical properties and morphology in tandem configuration are the other complementary factors contributing to locust's ultra-high aerodynamic performance. Furthermore, these mechanical properties (e.g. flexural stiffness) are influenced by the haemolymph (blood) circulation through the venations that are cleverly spread over the entire wing surface in almost all flying insects [15–19], investigated here in detail.

However, due to the convoluted and complex process of biological deciphering [20–23], in our previous study [2,24], we have resorted to the alternative solutions such as mathematical geometry optimization of the digitally reconstructed locust wings to enhance their numerically obtained aerodynamic efficiency. Although improvements of 77% and 14% were achieved for the two- and three-dimensional wing geometries, respectively, the results remain disputed until a sound experimental wind tunnel-based particle image velocimetry (PIV) measurement is conducted on the fabricated wing prototypes to estimate their actual aerodynamic performance. In this regard, here we explore several novel combinations of fairly economical and accessible manufacturing techniques to fabricate artificial wing prototypes sharing maximum similarity with their digitized and natural counterparts. Hence, apart from the availability and affordability, the pivotal concerns surrounding our proposed manufacturing procedure focus on: precise forming of the wing corrugations (i.e. numerically proven as one of the important features responsible for the aerodynamic performance [2,24]) onto the fabricated prototypes, and an optimum wing morphology (thickness and contour) [25–29] to deliver improved mechanical properties such as natural frequency and flexural stiffness compared with their published peer models [30–34]. Hence, prior to the fabrication process, a comprehensive finite-element analysis (FEA) is performed on the digitally reconstructed wings to validate their mechanical stability, performance and manufacturing-worthiness numerically.

To conduct a thorough exploration of the available bioinspired wing manufacturing techniques, certain overly simplified methods [35–39] that may not focus on mass optimization and precise fabrication of corrugations are dismissed in a process of elimination (PoE). Although several advanced manufacturing technologies such as micro-electromechanical systems (MEMS) etching [40–42] and cast micromoulding [43,44] potentially deliver the required precision for this study, their limited availability dissatisfies our primarily set conditions of accessibility and cost efficiency that in turn flout the idea of mass producing expendable drone wings. Furthermore, widely used experimental fabrication methods offer distinct advantages that must be hybridized to deliver the required performance collectively. For instance, the affordability of a fused deposition modelling (FDM) based 3D printer [45–48] combined with the precision of a commonly available vacuum thermoforming machine [49,50] and a laser computer numerical control (CNC) cutter may potentially result in development of the cost-effective prototypes with desirable aesthetic and mechanical properties.

To summarize, this study presents several hybrid techniques to explore their potential application in realizing digitized models of bioinspired wings. Artificial wing prototypes are fabricated in particular to validate their numerically determined aerodynamic performance, experimentally. Furthermore, we measure the maximum deformation rate and flexural stiffness in the direction of chord and wingspan to investigate the influence of corrugations, wing thickness, membrane, and fabrication technique on the prototype's mechanical properties. Our wing prototypes are developed particularly to satisfy two

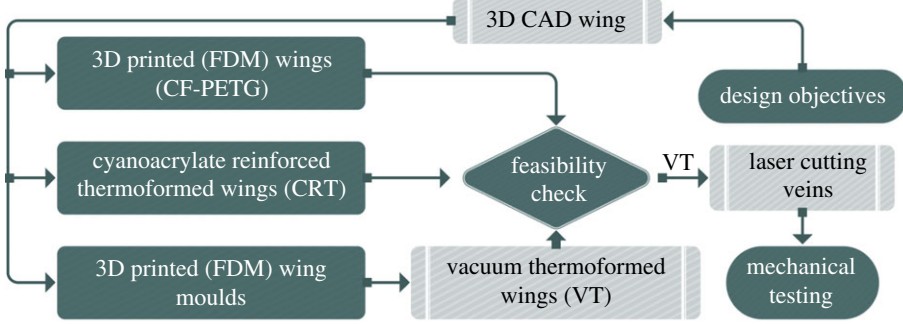

**Figure 1.** Flowchart representing the fabrication process flow from raw data input to the finalized wing prototype subjected to an experimental testing of maximum deformation rate and flexural stiffness.

pivotal objectives; maintain the numerically determined and optimized aerodynamic efficiencies (gliding ratio) from the previous study, and deliver a comparable or improved wing flexibility and resilience compared with that of a real locust's. Ultimately, a discussion on the practicality and mechanical properties obtained for different wing prototypes is provided to validate the proposal of the hybridized manufacturing procedure involving FDM 3D printing, vacuum thermoforming and low-power laser cutting to mass produce low-cost expendable flapping/gliding artificial wing prototypes that are light, thin and functional.

## 2. Material and methods

Initially, this section presents a clear overview of all the feasible (reasonably economical and accessible) manufacturing techniques explored in this study. In this regard, we employ a flowchart seen in figure 1 to paint a clear picture of the different fabrications' process flow presented. Since the processes are interconnected, the flowchart helps to track the progress of each procedure separately. First, the design objectives along with the three-dimensional computer-aided design (CAD) wing geometries from the digital reconstruction of the insect wing [2,24] are fed into our three proposed manufacturing techniques individually.

The foundation of these candidate methods are FDM 3D printing. It must be noted that the digital wings are scaled up by three times to remedy the resolution limitations of the 3D printing process. Fundamental properties such as weight, structural integrity and even aesthetics of the three output models are evaluated in a simple feasibility check for our application being micro aerial robots. The finalized prototypes, in this case, vacuum thermoformed (VT) wings pass the preliminary tests and therefore proceed with the laser CNC machining step involving trimming of its wing membranes using venation patterns. This stage further reduces weight and enhances structural stability and aesthetics. Ultimately, the prototypes are subjected to a comprehensive experimental evaluation of their mechanical properties involving maximum deformation rate and flexural stiffness to validate the proposed techniques through a performance comparison study.

### 2.1. Digital reconstruction

This study focuses on the gliding-expert insect, a desert locust (*Schistocerca gregaria*) shown in figure 2. Six active and healthy (undamaged wings) farm-bred adult female specimens were procured from an insect farm near Cangzhou, China. Taylor's method was followed in cementing the wing roots of the locusts in gliding posture using high viscosity cyanoacrylate [51]. Fixed wings were later sectioned at 20, 40, 60 and 80% spanwise chord, to be primed for geometrical digitization. Thickness of the fore- and hindwings were found to be variable due to venations ranging from 2 to 3 μm. The pseudo-microscopic scanning of the cross-section profiles were performed by carefully placing the sectioned wings in a lightbox fixed with a mirrorless Sony Alpha A6000 SLR camera armed with an AmScope 45× stereo microscope. Furthermore, magnified images of the wings in all three directions (top, side and front) were recorded before dissection in order to facilitate a precise three-dimensional digital reconstruction process involving micro sized details of the corrugations and fore-hindwing longitudinal & transverse separation. Ultimately, to form the three-dimensional wing structure in SolidWorks, a thin solid boss

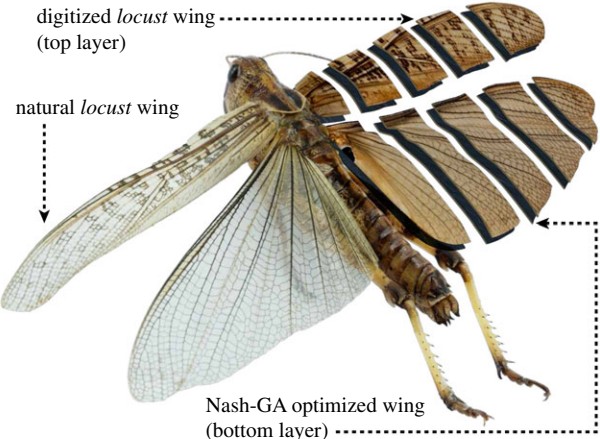

**Figure 2.** Graphical illustration of a desert locust (*Schistocerca gregaria*) with digitized wings sectioned at 20, 40, 60 and 80% spanwise chord, superimposed over its Nash-GA optimized counterpart from [2,24].

section was swept through the wing cross-section profiles at 20, 40, 60 and 80% spanwise chord, along the contour (guide curve) seen in figure 2 i.e. locust's wing planform. It must be noted that figure 2 is a hybrid illustration of the macroscopic view of wing cross-sections (aerofoils), CAD model and the natural insect's image merged into a single intuitive figure.

The digitized wings are numerically tested for aerodynamic performance in [2,24], and enhanced using Nash-GA hybrid optimization method resulting in two digitized wing designs whose aerodynamic performance shall be evaluated experimentally in the future study. Meanwhile, the CAD files of the wings are subjected to an advanced FEA to establish the manufacturing-worthiness of the designs which are then adapted for FDM additive manufacturing by designing bridges and overhang supports with minimum intrusions. Such additional bolstering structures are vital for 3D printing complex models with subtle cross-sections involving considerable amount of suspended layers (overhangs). Since this study proposes novel hybridization and configuration of two or three manufacturing methods, we present the overall fabrication process in two subsequent stages involving; moulding (preparation of wing moulds), and casting process that is further followed by two distinct curing processes.

## 2.2. 3D printed forewings

Before advancing with the hybridization of different manufacturing techniques, we identify the shortcomings of the already available procedures by exploring an end-to-end additive manufacturing (3D printer) based prototyping of bioinspired artificial wings frequently proposed in the literature [45–47]. Our main aim is to achieve similar if not enhanced prototypes with a lower grade of equipment. For instance, instead of hiring an experimental high-resolution 3D printer used in the literature (e.g. Object EDEN260V), we rather explore full potentials of a desktop consumer-grade 3D printing machine to achieve our objectives. This is possible through experimentation with numerous parameters defining the entire printing process. Some of these parameters include infill rate, extrusion and travel speed, layer height, and bed/extruder temperature, etc. Additionally, printer environment (ventilation, heating) is critical in maintaining an almost ideal condition for printing a complex CAD model flawlessly. Detailed description of the rapid prototyping equipment and settings are provided in the following section. Meanwhile, the artificial wing (CF-PETG-1) seen in figure 3 is a sample printer-adapted prototype that is composed of a wing frame enclosing locust-inspired major venations. To compensate for our device's moderate printing resolution (150 μm), layers are deposited with the wing planform parallel to the bed's XZ-plane.

Initially, we use a biodegradable polylactic acid (PLA) filament, which is one of the most printer-friendly materials with minimum warping and low melting point, to evaluate the maximum achievable precision in producing corrugated venations on the available printer. Additionally, to extend the scope of this section, we experiment with an unconventional material that is a fusion of carbon fibres and polyethylene terephthalate glycol (CF-PETG) to obtain a reasonably lighter and resilient prototype more similar to the published samples for a less biased comparative study.

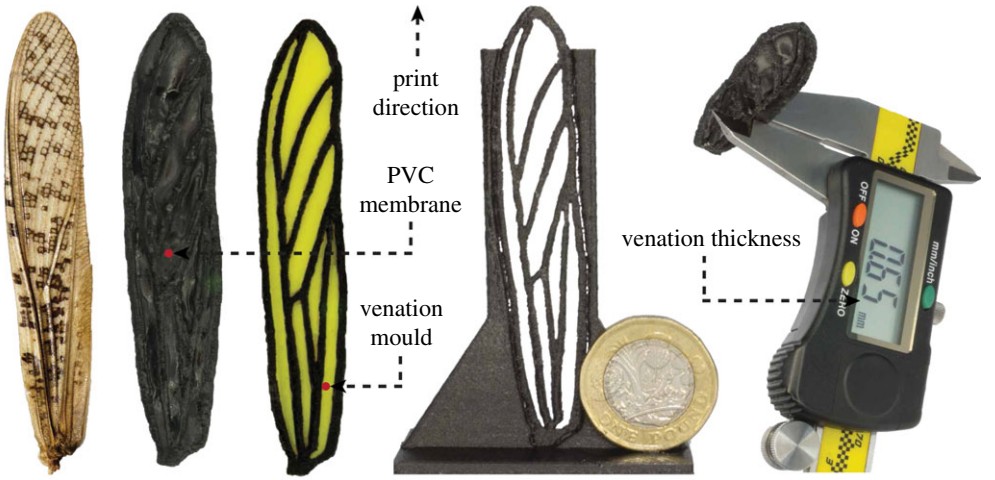

**Figure 3.** 3D printed locust-inspired forewing preparation process from left to right: actual locust forewing, finalized 50 μm PVC reinforced forewing prototype (CF-PETG-1), venation pattern mould, 3D printed forewing exoskeleton, and the measured average profile thickness.

Ultimately, the printed prototypes (venations of 650 μm) are post-processed by carefully filing away surface residuals using alcohol and fine-grit sandpaper. The exoskeletal wing structure (venations) shown in figure 3, is reinforced with a 50 μm PVC membrane (i.e. commonly available as vinyl wrap) that is further sprayed with a fluid thermoplastic rubber dissolved in a plasticizer solution composed of 5% methyl ethyl ketone, 15% toluene, 15% hexane, 30% resins and 35% VM & P Naphtha. This layer helps improve the prototypes' resilience against fatigue failure. Our proposed novel combination of thin PVC membrane with plasticized rubber delivers improved balance of flexibility and stiffness when compared with the membranes described in the literature [47,52] involving a subtle layer of brittle chitosan (linear polysaccharide 1,4-linked 2amino-deoxy-b-D-glucan) that is much similar to a delicate flying insect wing membrane. Detailed comparative study of the presented prototype is provide in §4

## 2.3. Rapid prototyping of wing moulds

Here, we provide a detailed description of the additive manufacturing procedure proposed for this study. Firstly, to satisfy the primary objective of maintaining cost-efficiency and accessibility, we hire an intermediate-level FDM 3D printer (Taz5 LulzBot, Aleph Objects, Inc. CO, USA) with a single extruder and a 0.25 mm stainless nozzle operating on a custom slicing software (Cura LulzBot Edition 3.6.18). Refinement of nozzle head from 0.5 to 0.25 mm is necessary to maintain precision while printing the subtle corrugations without compromising rigidity. Since the casting process involves elevated temperatures, achieving adequate thermal resistance and robustness is crucial for developing the perfect wing mould. Apart from printer settings and model adaptions, filament material plays a vital role in tackling the challenges associated with the rapid prototyping of complex CAD models. A correct choice of printing material can easily address the overhang problems, surface fineness, warping, etc.

This can be achieved by conducting a comprehensive process of trial and error, experimenting with a range of commonly available filaments shown in figure 4, including a fluorescent yellow polyethylene terephthalate glycol (PETG), blue polycarbonate (PC), red acrylonitrile butadiene styrene (ABS), and a yellow polylactic acid (PLA) material. Quantitative properties of the above materials are tabulated in the table 1, to identify each of their advantageous characteristics. Visual observation of the printed wing moulds on the other hand, provide further useful information validating the tabulated data (from Simplify3D). For instance, the PC hindwing mould with maximum durability and strength offers the poorest aesthetics due to its lower printability rate.

Although each material exhibits a certain desirable property, a fusion of one or two materials could potentially result in a single filament with all the required characteristics collectively. Shown on the printer bed in figure 4, the grey mould (CF-PETG) is a result of such hybridization where the thermal stability and specific strength of carbon fibre (CF) is fused with the flexibility and high printability

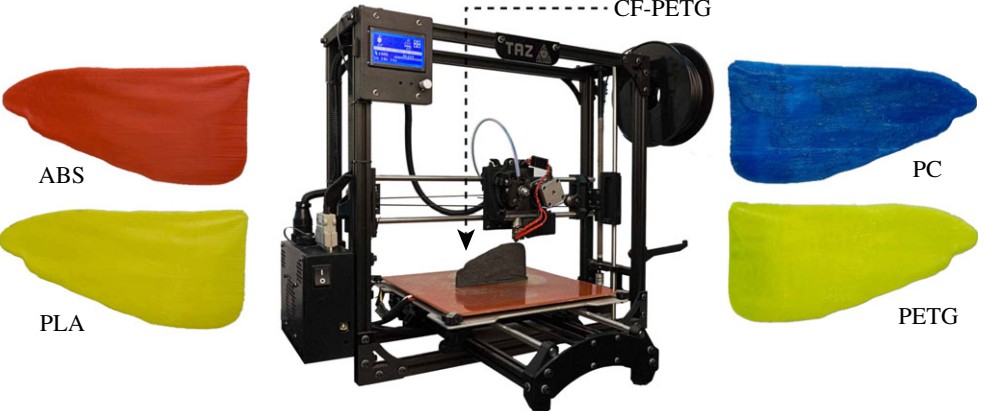

**Figure 4.** LulzBot Taz5 machine additively manufacturing hindwing moulds using various commonly available materials such as PETG, PC, ABS, PLA and carbon fibre-infused PETG (CF-PETG) on printer bed.

rate of the PETG filament. It must be noted that the figure 4 illustrates only the hindwing moulds due to the following sections' focus on forewings mainly consistent with the literature. Post-processing of the moulds involved surface filing of the minor roughness caused due to additive layer deposition. However, to prevent over-grinding and tarnishing of the precise corrugation profiles, only fine-grain sandpapers were used with incremental grits from 500 to 2000.

## 2.4. Cyanoacrylate reinforced thermoforming

As the next stage of the fabrication process, we advance with the casting phase involving experimentation with different techniques to emphasize the significance of this stage in achieving the desired corrugation profiles on the finished prototypes. First, a rather simplified method is proposed for this study involving thermoforming of a 50 µm plasticized polyvinyl chloride sheet (commonly used vinyl wrap) to act as the wing membrane reinforced with venations of cyanoacrylate. The process of thermoforming involves a 2000 W heat gun blasting a jet of hot air on the thermosetting self-adhesive PVC sheet reaching its pliable state. At elevated temperatures, a positive force is simultaneously applied uniformly over the PVC sheet to coat the entire surface of the wing moulds with the help of fine crafting tools. Although the thermoformed material is thermosetting (cures upon cooling), an accelerated crystallization of the polymers by freezing is preferred in this case to increase the robustness of the cured prototypes seen in figure 5b.

Rapid cooling improves the maximum elastic deformation capacity of the cured PVC. However, scaling up (3×) of the digitized wings, on the other hand, increases wingspan-to-thickness ratio (greater than 1000) which has an adverse effect on the rigidity and stiffness of prototyped wings. Therefore, a thin (approx. 3 mm) layer of high-grade viscous cyanoacrylate adhesive is manually injected through a needle over the moulded wings along the locust-inspired venations printed on the negative mould (red contours) as shown in figure 5b. Furthermore, as the second part of a two-stage curing process, sodium bicarbonate-based accelerator is sprayed to rapidly solidify the reinforcement layers of cyanoacrylate. Shown in figure 5c, the finalized CRT prototypes possess the desirable specific strength at reduced weight and venation thickness. Detailed comparative study of this prototype's mechanical properties is provided in §4.

## 2.5. Vacuum thermoforming

Ultimately, to maintain consistency in satisfying the primary objectives of accessibility and affordability, this study explores the widely used forming process called vacuum thermoforming (which commonly produces disposable containers and packaging) to evaluate its potential application in the aerospace industry for mass production of expendable drone parts. The workshop for this manufacturing method is mostly hired for industrial purposes with high-volume orders on tight operation schedules. Therefore, a local contractor with a limited production capacity agreed to dedicate their facility for several hours along with technicians to assist the thermoforming of our artificial wing prototypes as a contribution to science. Initially, one side of the 3D printed wing moulds were flattened with

**Table 1.** Material properties of different 3D printed wing moulds.

| mould printed material | maximum resolution (μm) | minimum profile thick (μm) | mould density (g cm$^{-3}$) | stiffness rate (%) | coefficient thermal expansion (μm m$^{-1}$ °C$^{-1}$) | ultimate strength (MPa) | durability rate (%) | printability rate (%) |
|---|---|---|---|---|---|---|---|---|
| PLA | 140 | 250 | 1.24 | 75 | 68 | 65 | 40 | 90 |
| ABS | 150 | 250 | 1.04 | 50 | 90 | 40 | 80 | 80 |
| PC | 250 | 500 | 1.2 | 60 | 69 | 72 | 100 | 60 |
| PETG | 150 | 250 | 1.23 | 50 | 60 | 53 | 80 | 90 |
| CF-PETG | 180 | 250 | 1.3 | 100 | 57.5 | 68 | 30 | 80 |

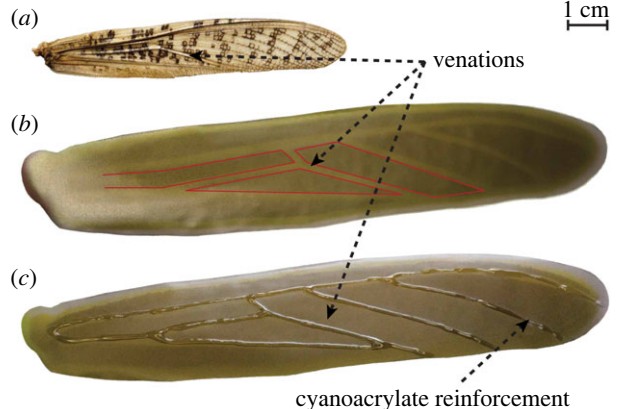

**Figure 5.** Cyanoacrylate reinforced thermoforming; (*a*) scanned real locust forewing, (*b*) 50 µm PVC sheet cast on a 3D printed corrugated wing mould with negative venations, (*c*) finalized prototype with cyanoacrylate reinforcement along venations.

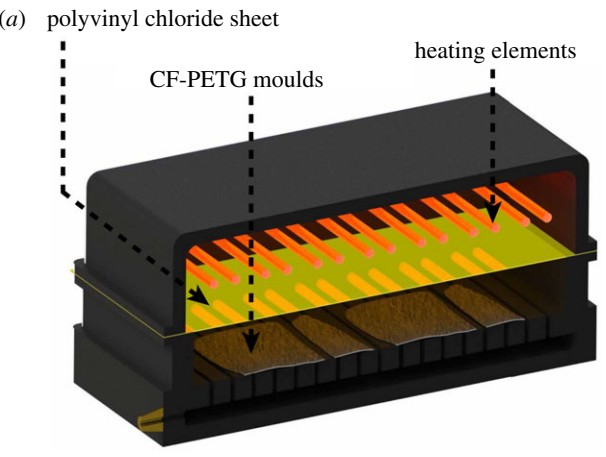

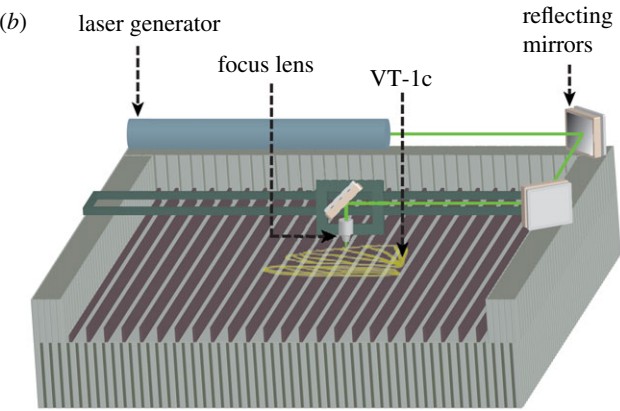

**Figure 6.** Final manufacturing of wing prototypes; (*a*) vacuum thermoforming of the PVC sheets, and (*b*) laser cutting of the membranes through venation patterns.

modelling clay to be prostrated over a perforated steel tray custom-made for our moulds. As seen in figure 6*a*, the tray is placed inside the oven that is heated up to PVC's pliable temperature (130 ± 10°) facilitating an accurate forming of moulded corrugations onto the prototypes upon exertion of negative pressure from the vacuum chamber underneath the moulds. Ideally, a CNC machined aluminium mould is used for this fabrication technique to endure its heat and pressure-based forming. However, our limited production volume and high precision requirements inspired the

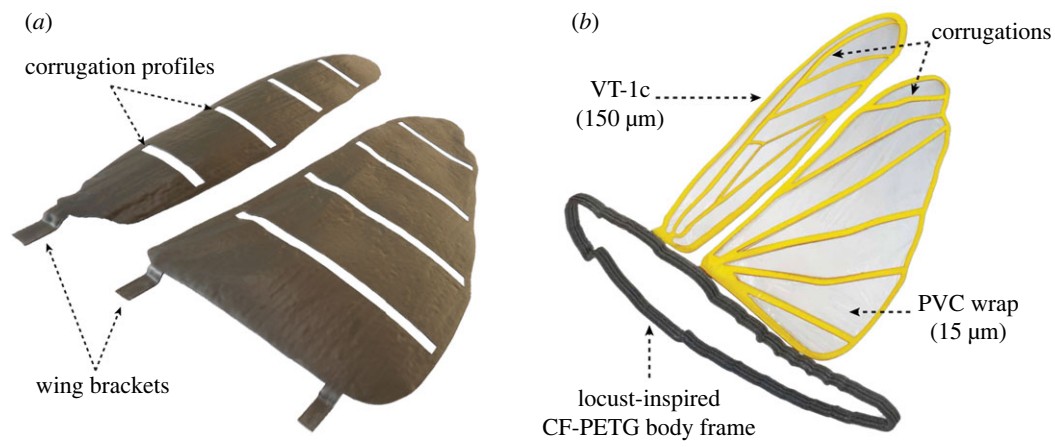

**Figure 7.** Illustration of the vacuum thermoformed PVC wing prototypes; (*a*) VT-1s prototype i.e. a VT-1 wing sectioned at 20%, 40%, 60% and 80% spanwise chord to demonstrate the obtained corrugations, and (*b*) laser-trimmed 150 µm thick artificial fore- and hindwing (VT-1c) attached to a locust-inspired CF-PETG printed body frame.

experimental fabrication of the most suitable 3D printed CF-PETG based mould that offered sufficiently high thermal resistance and stiffness to serve the purpose at reduced production cost.

Post-processing involves $CO_2$ laser CNC trimming of the prototype membrane along wing planform and venation patterns shown in figure 6*b*. Notable advantages of this cutting method include: reduced labour and costs, high precision, accessibility, ease of operation and repeatability. Since PVC material is sensitive to extreme temperatures, with the help of a trial and error on the unformed PVC sheets, the laser temperature and speed are calibrated for each wing thickness to avoid overheating, melting, annihilation and inaccurate criss-cross cutting paths. A VT-1 prototype is sectioned at 20, 40, 60 and 80% spanwise chord to obtain VT-1s shown in figure 7*a* emphasizing the precision obtained in curing of the wing corrugations onto our final prototypes. We present this case in order to illustrate the significance and influence of the obtained wing corrugations on the structural performance of the prototypes as mentioned in §2.3. Additionally, a non-corrugated (unformed) PVC sheet called VT-1p is laser trimmed according to the planform and venations of a locust wing to be tested in the next section for flexural stiffness. On the other hand, the 150 µm thick fore- and hindwing (VT-1c) mounted on a locust-inspired CF-PETG printed body frame shown in figure 7*b*, is considered as the finest prototype proposed in this paper. Furthermore, several distinct model variants for each fabrication technique is developed by implementing PVC sheets of varying thickness (150–300 µm) to highlight the influence of this parameter on the mechanical properties of the prototypes developed.

# 3. Mechanical properties testing

Aerodynamic performance of insect wings are directly associated with their mechanical properties. Numerous factors contribute and influence these characteristics. One such influential element that is recognized in the literature since the mid-eighteenth century [15–19] is a fluid called haemolymph (analogous to the blood in vertebrates) flowing through the insect veins that are extended throughout its body, particularly covering its inner wing surface. Here, a collection of common flying insect species are considered in this study to analyse the precise effects of this fluid on the wing flexibility of invertebrates. Other properties such as wing flexion also play an important role in the manoeuvrability of flying insects. However, due to a lack of sufficient data, it is not possible to conclude whether stiffness and flexibility are beneficial in their absolute forms on an insect's wing. Perhaps a balanced trade-off strategy must be maintained in order to achieve all the objectives such as weight reduction, resilience and agility all at once. This is reflected in the presented study that avoids recommending a one-size-fits-all material or structure for flapping micro aerial vehicle (MAV) applications. Having mentioned that, we evaluate the flexural stiffness and maximum deformation rate in chordwise and spanwise direction of the wing prototypes both numerically (FEA) and experimentally. These properties, tabulated in table 2, reveal the significance and influence of the fabrication technique, material used, and wingspan-to-thickness ratio, on each distinct prototype developed in this study. In total, seven different samples were evaluated as: (i) VT-1p that is a non-

**Table 2.** Mechanical properties of different natural and artificial wing prototypes.

| wing prototype | wing mass (mg) | profile thickness (μm) | span (mm) | chord (mm) | flexural stiffness (Nm$^2$) spanwise | chordwise |
|---|---|---|---|---|---|---|
| VT-1p | 376 | 150 | 146 | 27.8 | $5.67 \times 10^{-5}$ | $2.96 \times 10^{-6}$ |
| VT-1c | 667 | 150 | 144 | 28.3 | $1.34 \times 10^{-4}$ | $5.67 \times 10^{-6}$ |
| VT-1 | 2019 | 150 | 148.9 | 28.5 | $1.27 \times 10^{-3}$ | $4.49 \times 10^{-4}$ |
| VT-2 | 2085 | 150 | 149.8 | 28.2 | $1.70 \times 10^{-3}$ | $4.82 \times 10^{-4}$ |
| CF-PETG-1 | 773 | 650 | 74.23 | 13.78 | $7.30 \times 10^{-4}$ | $3.66 \times 10^{-5}$ |
| CF-PETG-2 | 2700 | 1500 | 147.5 | 28.1 | $4.78 \times 10^{-3}$ | $8.21 \times 10^{-5}$ |
| CRT | 737 | 150 | 148.7 | 27.15 | $2.96 \times 10^{-4}$ | $4.34 \times 10^{-6}$ |
| cicada [53] | 17.6 | $70 \sim 720$ | $\simeq 39$ | 12 | $1.03 \times 10^{-5}$ | $6.65 \times 10^{-7}$ |
| artificial cicada [53] | $10.38 \sim 24.04$ | $71 \sim 221$ | 39 | 12.5 | $2.18 \times 10^{-5} \sim$ $8.91 \times 10^{-7}$ | $8.34 \times 10^{-7} \sim$ $1.5 \times 10^{-7}$ |
| M. sexta [54] | 35.76 | $30 \sim 500$ | 49.9 | 17.8 | $2.09 \times 10^{-5}$ | — |
| artificial M. sexta [54] | 80 | $100 \sim 200$ | 50.6 | 20.6 | $1.69 \times 10^{-4}$ | — |

corrugated (unformed) 150 μm PVC wing, (ii–iv) VT-1c,1,2 being laser cut 150 μm and the uncut 150 and 300 μm VT PVC wings, (v–vi) CF-PETG-1,2 being the initially 3D printed prototypes with varying membrane and scaling factor, and (vii) CRT, the thermoformed 50 μm PVC wing with cyanoacrylate reinforced venations.

## 3.1. Effects of haemolymph on insect wing stiffness

Earliest documented evidence of research on the haemolymph circulation across insect wing venations dates back to 1744 [15], when Henry Baker reported his observations on the fluid flow through a grasshopper's wing (a member of the locust family). Today, with the development of this research topic [16–19,55], there is sufficient evidence bolstering the claim that this phenomenon occurs in almost every winged insect to different degrees and it is responsible for both pressurization as well as moisturization of the wing cuticles achieving flexion and flexibility, respectively. This paper also, further develops on the topic to identify the required characteristics of a perfect bioinspired artificial wing. Although a generalized conclusion cannot be drawn on whether or not a haemolymph flow in the wing is purely advantageous, it is definitely worth studying its effects on the associated mechanical properties. This is the point of relation between this section and the overall objective of the presented study (i.e. development of an efficient bioinspired wing). However, since the realization of an artificial wing with veins and fluid flowing through them is practically not useful with the current technological development, we need to identify the simplest yet most effective natural wing design in the meanwhile. Here, simple implies least dependence on haemolymph flow. In this regard, we investigate a pivotal mechanical property (flexural stiffness) associated with the wings of a number of common flying insects shown in figure 8a namely, a desert locust (Schistocerca gregaria), black African beetle (Heteronychus arator), dragonfly (Anisoptera), Bi-ab cricket—a rare winged variant of spiked magician (Saga pedo) named after the place it is found in abundance, brown house moth (Hofmannophila pseudospretella), and a housefly (Musca domestica). Detailed test set-up and the calculation of the flexural stiffness is demonstrated in §3.3. Insect wings were primed for experimentation in two configurations: (i) attached to the body with haemolymph flowing through but fixed with cyanoacrylate at the root representing a beam structure, and (ii) detached and dried for up to 24 h at $25 \pm 2°$ and $20 \pm 5\%$ humidity. Figure 8b clearly demonstrates the level of influence haemolymph has on each insect wing flexibility. The plots in blue and red represent flexural stiffness determined for each insect's wing in configuration (i) and (ii), respectively. The graph is informative in terms of identifying the insect species whose wing flexibility has minimum dependence on its haemolymph flow that in turn indicates its suitability for inspiring

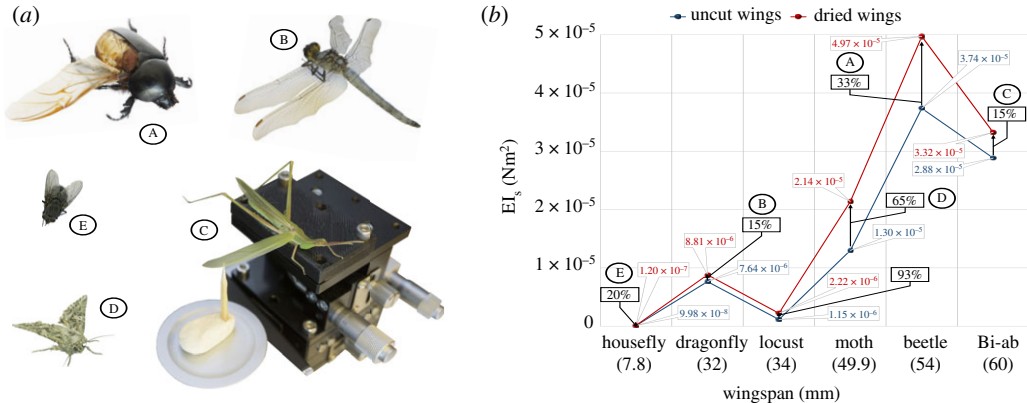

**Figure 8.** Illustration of the influence of haemolymph on mechanical properties of wings; (*a*) flexural stiffness test performed on the wings of insects labelled as (A) black African beetle (*Heteronychus arator*), (B) dragonfly (*Anisoptera*), (C) Bi-ab cricket, (D) brown house moth (*Hofmannophila pseudospretella*), (E) housefly (*Musca domestica*), and (*b*) graphical plot of the flexural stiffness obtained for the insect wings in two configurations namely uncut (blue plots) and dried wings (red plots).

artificial wing designs. Considering the wingspan and area, Bi-ab cricket offers the most independence, further proving them as a better source of inspiration for future studies. Although this insect belongs to the locust family, its wings are proven simpler to mimic when compared with the *Schistocerca gregaria*. The results demonstrated here can be simply validated by visual observation of the thickness and density of venations on the wings. For instance, softer wings on the locust and moth (D) seen in figure 8*a* upon dessication of their valuable haemolymph, demonstrate the largest relative increase in wing stiffness being 93% and 65%, respectively. This is considered as a challenge to the biomimetic researchers focusing mainly on dragonfly and locusts as a source of inspiration to design bioinspired MAVs. Although there remain a multitude of intriguing and unanswered questions related to this topic, such as the real reason behind the natural wings' haemolymph content, true or perhaps concealed purpose of this fluid flow through wing venations, its potential advantages for an MAV wing application etc., we propose these as a potential topic for future study.

## 3.2. Finite-element analysis

The three-dimensional digitized model of the locust tandem wing [2,24] used above for the fabrication processes, was initially subjected to a comprehensive numerical analysis of its mechanical performance in ANSYS Workbench to establish its manufacturing-worthiness. In other words, each design must pass this stage to be ready for the fabrication process. Since the material properties and minimum manufacturable thickness is defined in the numerical tests, a design failing to satisfy these conditions must undergo an adaptation process. Primarily, the 150 μm thick geometry is meshed as seen in figure 9. It is crucial to generate as much structured grids as possible to increase the solution accuracy and computation performance of the analysis. In this case, the heavily refined meshed body consisting of 600 thousand hexagonal cells sized at $5 \times 10^{-4}$ m produced satisfactory solution with no significant change (less than 5%) in the computed deformations upon further refinement. Later, boundary conditions were defined by constraining the wing roots and leading-edge to a single rotational degree of freedom (DOF) about the *X*- and *Y*-axis for the estimation of flexural stiffness chordwise and spanwise, respectively. Since the axilla that is responsible for three-dimensional motion of the wings was not considered in our model, we ignore the other two DOFs. Finally, a point force (structural load) equivalent to the aerodynamic forces acting on the real locust wing as shown in figure 9, was uniformly applied at almost 70% of the wingspan and chord for the estimation of flexural stiffness spanwise (blue) and chordwise (red), respectively. Also a fixed constraint at the wing root (blue lock) and leading edge face (red locks) of the wing was applied for the EI calculation in spanwise and chordwise, respectively.

The static structural solution of our proposed model involved a set of quantitative and qualitative results demonstrating the directional and total deformation, and equivalent (von-Mises) stress/strain distribution. As an example, figure 10 shows the equivalent (von-Mises) stress distribution over the simulated wing prototypes VT-1(FEA) and VT-1c(FEA). It is evident that the stress concentration at the wing roots has expanded as a result of removal of wing membrane along venation pattern; however, the maximum stress experienced by VT-1c(FEA) is yet to exceed midway through the safety

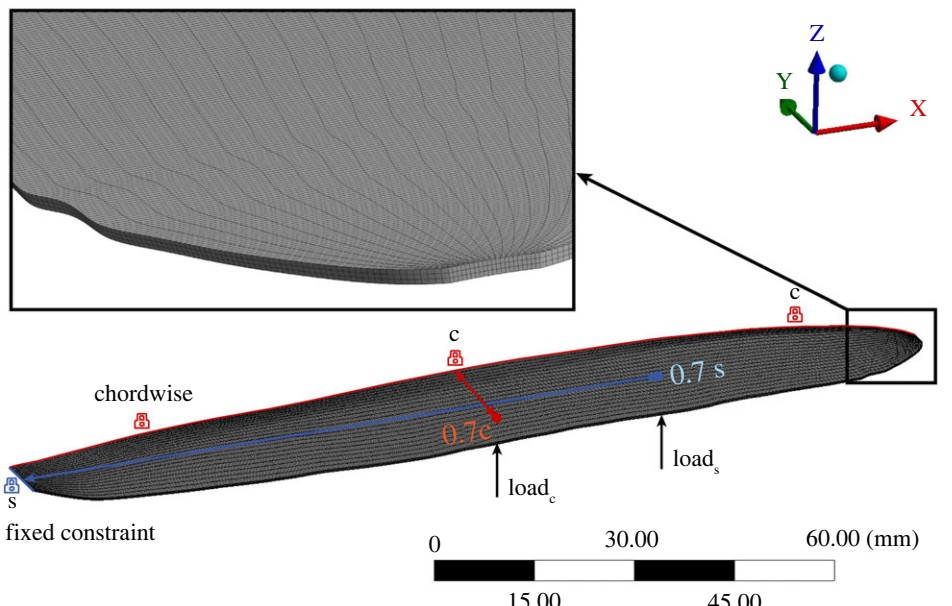

**Figure 9.** Illustration of the final three-dimensional meshing of the digitized locust forewing with the magnified representation of the structured grid formation near its wing tip.

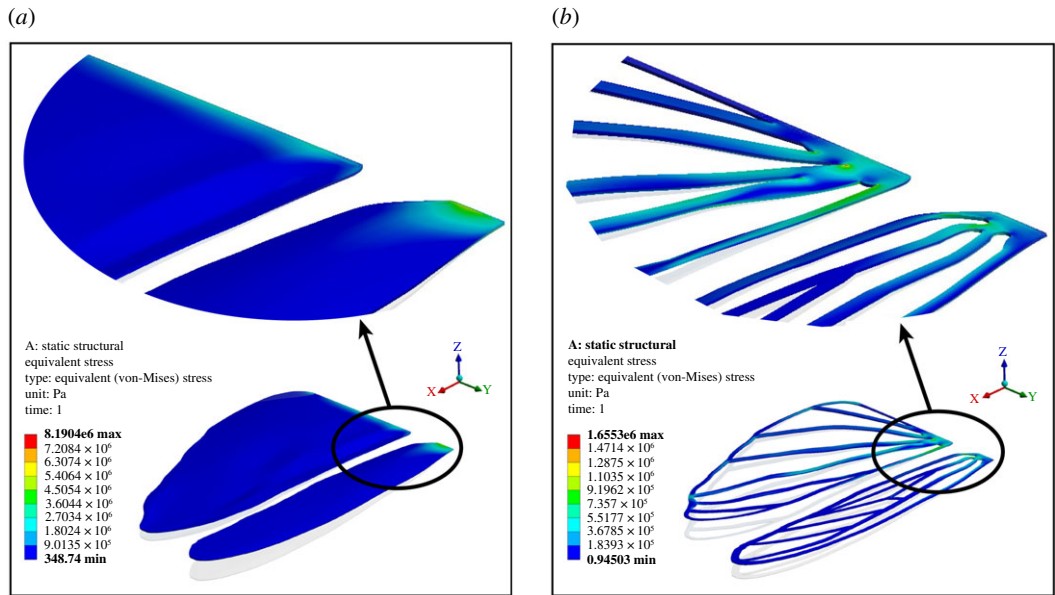

**Figure 10.** Illustration of the equivalent (von-Mises) stress distribution contour over the simulated locust-inspired tandem wings; (*a*) VT-1(FEA) and (*b*) VT-1c(FEA).

factor. Quantitative comparison of the obtained results is also plotted on the graph shown in figure 11*b*. The hollow and filled square markers represent VT-1c(FEA) and VT-1(FEA), respectively. A clear relationship between the experimental and numerical results is established as their differences are very negligible (round and square markers plotted closely) that further bolsters the reliability of the experimental set-up and the results obtained. Furthermore, the FEA analysis established the manufacturing-worthiness of the digitized wing models by demonstrating the fail-safe properties of delicate venations prior to 3D printing or laser cutting.

## 3.3. Flexural stiffness (EI)

As a fundamental mechanical property, flexural stiffness defines a wing's ratio of deformation to aerodynamic loading. This property is measured for our prototypes to validate their mechanical

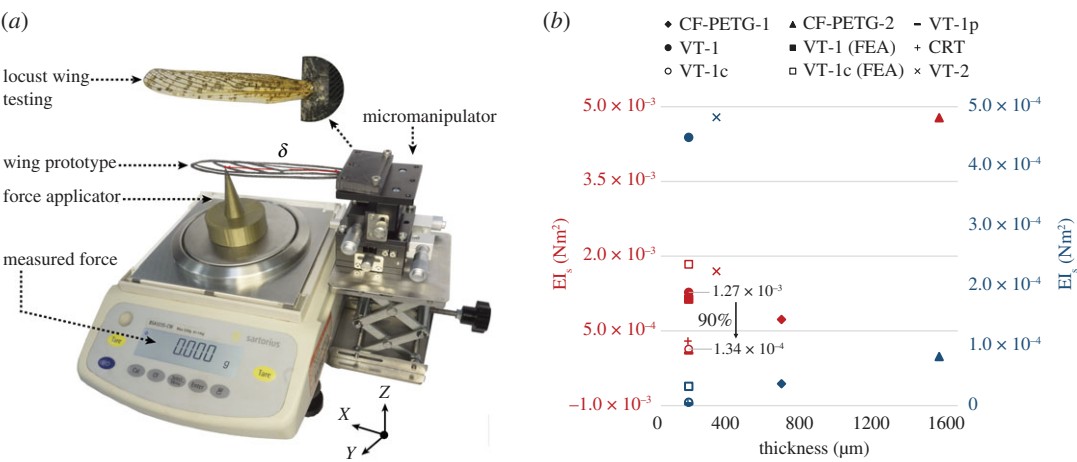

**Figure 11.** Flexural stiffness evaluation experiment; (*a*) test set-up comprising of an electronic microbalance and a three-axis micromanipulator measuring the EI for CF-PETG-2 forewing prototype along with a magnified image of the real locust forewing testing, and (*b*) plot of the flexural stiffness obtained spanwise (red) and chordwise (blue) for different wing prototypes with varying thickness.

performance with respect to their peer models from the literature. Structural analyses are usually categorized as destructive and non-destructive, where the latter, such as a flexural stiffness measurement, is prioritized due to their test-subject reusability. Following the standard test set-up and procedure from the literature [56,57], this mechanical property is measured mainly with the help of two instruments, namely, a three-dimensional manual micromanipulator (Shengling, LD60-RM, China) with 10 µm resolution, and an electronic balance (Sartorius BSA323S-CW, China) with 1 mg precision shown in figure 11*a*. Since the micromanipulator is a right-hand version (RM), the wing prototypes are root clamped to its left edge with their chord parallel to the instrument's XY-plane.

Here, the wing aerodynamic loads are represented by a relatively sharp pin placed vertically on the electronic microbalance exerting a point force on the wing prototype at 70% of the chord and wingspan for the measurement of flexural stiffness in chordwise and spanwise direction, respectively. Anticlockwise rotation of the manipulator's z-axis micrometer lowers the wing towards force pin at $\delta$ mm from leading-edge and clamped-edge for spanwise and chordwise measurements, respectively. This parameter is clearly indicated using red lines on the wing being tested in figure 11*a*. It must be noted that the wing displacement ($\Delta$) measured on the z-axis micrometer is considered from the point a force reading ($f$) appears on the microbalance (i.e. when the test subject comes in contact with the pin). Using the recorded readings, the flexural stiffness [31,56] is measured as, $EI = (f\delta^3/3\Delta)$ where displacement on the wing ($\Delta$) is maintained at 5% of $\delta$ to associate a minor deformation rate with the flexural stiffness.

The measured EI spanwise and chordwise for all the prototypes is tabulated in table 2, and plotted against their profile thickness in figure 11*b*. The graphical plot comprises nine different markers (square, triangle, circle, etc.) representing each tested model described in the top legend. Furthermore, markers are categorized in two colours of blue and red indicating the direction of EI measurement in chordwise ($EI_c$) and spanwise ($EI_s$), respectively. Although a detailed discussion on the results is provided in §4, a rapid observation of the graph suggests direct proportionality of the measured rigidity to profile thickness. Therefore, our thinnest (150 µm) prototype exhibits double the stiffness of a natural locust wing, which is three times thinner. Additionally, the spanwise flexural stiffness is found to be greater than its chordwise estimation by an order of a magnitude which validates our experimental set-up and configuration according to the literature [31,56]. This difference in spanwise versus chordwise stiffness is a standard phenomenon observed in natural wings as well. However, two most significant findings of this analysis are to demonstrate the influence/advantages of forming corrugations onto the wings, and the process of membrane removal through laser cutting venations, on the flexural stiffness and weight of the VT-1c prototype that are considerably improved by 90% and 67%, respectively.

## 3.4. Maximum deformation rate

Maximum deformation rate is the measure of a structure's capacity to resist a fracture failure. The test platform for this parameter shown in figure 12, is almost similar to the previous set-up in principle. The prototyped wings are again attached to the left edge of the micromanipulator translating in Z-plane to

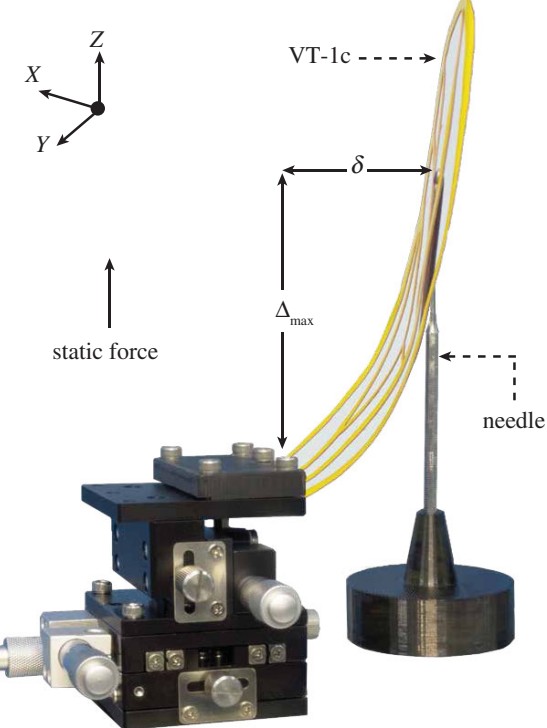

**Figure 12.** Illustration of the maximum deformation rate measurement set-up, VT-1c undergoing a large deformation (greater than 148.2%) without fracture.

apply a point force representing aerodynamic loads at 70% of the chord and wingspan for the measurement of maximum deformation rate chordwise and spanwise, respectively. Estimation of the two mechanical properties presented in this study are fundamentally different only in terms of the wing displacement rate ($\Delta$) which was previously maintained within 5%. However, this is increased substantially above 5% until a visually evident fracture occurs on the wing surface, indicating its maximum deformation rate. Interestingly, the prototypes developed in this study offer much larger range of plastic deformation in contrast to the brittle CF-based peer models from the literature [53], i.e. our proposed artificial wings resist immediate fracture upon yielding. As an example, the VT-1 prototype remains intact even at a massive wing displacement of $\Delta_{max} = 100$ mm that suggests an almost flawless deformation rate of greater than 97.6%, which is double the reported 57.9% obtained for an artificial *cicada* wing [53]. The laser-cut prototype (VT-1c), on the other hand, delivers a further improved performance (virtually indestructible) compared with the other models in this study, due to its reduced weight and stiffness recording an incomparable maximum deformation rate of greater than 148.2%, as seen in figure 12. Such reliable mechanical performance is particularly favourable in aerospace industry due to the higher standard requirements defined by the airworthiness of an aerial vehicle.

# 4. Discussion

In this section, mechanical properties of only the selected wing prototypes that pass the preliminary evaluations involving aesthetics and consistency, are investigated in detail. The significance of corrugations are emphasized in this stage, where the unformed VT-1p or the CRT prototype fail to pass the consistency test. As seen in table 2 and figure 11*b*, the EI for VT-1p is more comparable to that of the real locust wing's stiffness, and the CRT's performance is very similar to that of the VT-1c. However, both prototypes are overly flexible and fluid in nature that is suspected to be due to a lack of precise corrugations that may provide structural integrity withstanding their own weight against gravity. This does not imply that a real insect wing is overly flexible since the body mass and size of an insect such as a locust is not comparable to the readily available micro aerial robots. The thermoformed (VT) and 3D printed wings (CF-PETG) on the other hand, pass the visual qualitative analysis and are therefore subjected to a detailed quantitative evaluation of their flexibility. This is an

important mechanical property for a wing due to its continuous exposure to compressive, torsional and bending deformations while airborne. Flexibility is estimated by measuring the flexural stiffness and maximum deformation rate of a wing in chordwise as well as spanwise direction. These properties are measured and tabulated in table 2, suggesting their favourable performance aeronautically when compared with their peer models from literature [27,44,47,53]. This is speculated to be resulting from the main objective of the published models being focused on the biomimicry of an insect's mechanical and aesthetic properties, rather than aiming for an aerodynamically functional and pragmatic artificial wing prototype. From figure 11$b$, it can be inferred that the flexural stiffness is directly proportional to the wing thickness, size and density. Therefore, doubling the VT-1 thickness to 300 μm producing VT-2 increases both $EI_s$ and $EI_c$ emphasizing the significance of wing thickness in defining rigidity.

However, our finest prototype, VT-1c offers the highest flexibility both in chordwise as well as spanwise direction without compromising its sturdiness. Considering the large maximum deformation rate (greater than 148.2%) achieved on VT-1c, this lighter and thinner prototype is more recommendable for further aerodynamic performance evaluation via PIV measurements, as it also maintains the most similarity in attributes such as corrugations, planform and thickness when compared with its counterpart in nature. Additionally, these purpose-made wings are capable of withstanding flow-induced deformations in order to maintain their original geometry (optimized and unoptimized corrugations) for an effective aerodynamic performance evaluation presented in the future study. It is important to note that the fabricated wings in this research are scaled by 3× resulting in an increase in their weight, which implies that they are still lighter compared with their peer models that are much smaller and impractical. The scaling is necessary for the intended application of the wings being readily available micro flapping wing robots. Therefore, we can conclude that the hybridization of an affordable and accessible 3D printing (additive manufacturing), vacuum thermoforming, and laser CNC cutting could potentially be hired for future mass production of expendable low-cost drone wings or other fuselage parts.

# 5. Conclusion

This study explores several affordable and easily accessible manufacturing methods to propose a hybridized procedure capable of developing bioinspired artificial wing prototypes for micro aerial robots that are cost-effective and expendable. Particularly, a gliding-expert insect's (locust) wings are digitized and fabricated in distinct forms to evaluate the influence of all the pivotal factors such as profile thickness, material and corrugations on their mechanical properties. It can be concluded that the hybridization of additive manufacturing, vacuum thermoforming and laser cutting can result in the fabrication of a pragmatic 0.667g artificial wing prototype (VT-1c) scaled by 3× delivering beyond satisfactory mechanical performance quantitatively defined by a relatively balanced flexural stiffness ($EI_s = 1.34 \times 10^{-4}$ and $EI_c = 5.67 \times 10^{-6}$), and an incomparable maximum deformation rate of greater than 148.2%, i.e. elastically deformable past 90° bending. In addition, we present a comprehensive discussion and comparison of the qualitative and quantitative data highlighting the influence of wingspan-to-thickness ratio, material and casting method on each prototype's mechanical properties. Ultimately, extension of this research focuses on a novel computer-associated design technology to generate further mechanically optimized artificial wing concepts [58]. For the future developments of this area of research, further improvements in fabrication simplicity, reduction in size and weight is highly recommended. In particular, a deeper focus on further wing geometry optimization and enhancement of aerodynamic efficiency may facilitate successful achievement of improved prototypes more suitable for the endured flight applications.

Ethics. We were not required to complete an ethical assessment prior to conducting this research.

Data accessibility. Our data are included in the electronic supplementary material.

Authors' contributions. H.I. and S.Y. designed the study and performed the finite-element analysis, drafted and critically revised the manuscript; H.I and C.X. performed the process of adapting CAD models to additive manufacturing, and performed the laser cutting; H.I., S.Y. and W.C. conducted the 3D printing and vacuum thermoforming process. All authors gave final approval for publication.

Competing interests. We declare we have no competing interests.

Funding. This work was supported by EU FP7 projects LIVCODE (grant no. 295151); HAZCEPT (grant no. 318907); EU Horizon 2020 projects STEP2DYNA (grant no. 691154); ULTRACEPT (grant no. 778062); Google Cloud Platform research credits, and partially supported by the National Natural Science Foundation of China (grant no. 91648203); Natural Science Foundation of Hubei Province (grant no. 2018CFB431).

Acknowledgements. The authors thank the reviewers for their valuable comments.

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
