## [Peer Review File · Royal Society Open Science]

Review History

RSOS-202253.R0 (Original submission)

Review form: Reviewer 1

Is the manuscript scientifically sound in its present form?

No

Are the interpretations and conclusions justified by the results?

No

Is the language acceptable?

Yes

Do you have any ethical concerns with this paper?

No

Have you any concerns about statistical analyses in this paper?

No

Recommendation?

Reject

Comments to the Author(s)

The authors present a study investigating the manufacturing feasibility and mechanical performance of bio-inspired insect wings. The model is a desert locust and the selection of wings are made by 3D printing, vacuum thermoforming, and laser cutting, or a combination of these methods.

The study refers to the authors' previous work using a Nash algorithm to optimise wing geometry beyond that seen in Nature given a limited set of objectives, although the link is not clear to the present work, the vein pattern is a reduced version of that found in the locust, scaled three fold. In a step towards testing the aerodynamic performance of models experimentally, and subsequently adding them to a glider, they explore several manufacturing techniques to identify the most appropriate against the criteria of accessibility, cost and structural performance. Aerodynamic tests and glider construction are not investigated here.

The manuscript describes a substantial amount of work. However, two sections seem rather out of place. There is a section on computational modelling which, while needing a little more methodological detail, seems to be reasonable. However, I did not see how it added to the story because the physical tests on prototypes are not used to validate the simulation, and they are also more comprehensive. There is also a section regarding haemolymph, but this does not seem to form part of the bio-inspired wing objective. These two sections disrupt the flow of testing wings made by different designs and manufacturing processes.

The manuscript concludes that one of the fabrication methods is better than the others but goes further to say it has "optimal" stiffness properties, which is not well supported.

I would invite the authors to consider what material properties would be ideally required for a manufactured glider (or flapping MAV). Specifically, what stiffness is optimal? This might well be different from the insect it mimics. Is flexion of the wing adaptive/beneficial in some way, or is the buckling simply a product of looking to minimise material/weight? If the latter, then would a stiffer wing be better? Or could a softer wing be ok too?

What is the material effect of haemolymph? Does it make a direct contribution to the bending properties (e.g. by pressurising the system) or is it an indirect effect, (e.g. by preventing desiccation of the cuticle)?

Because there are several fabrication methods, it is sometimes difficult to follow each through the workflow from the insect model start to engineered finish.

Specific comments:

"Superior structural performance of locusts." Measured against / superior to what?

L22. Veins do not 'coat' the wing surface.

P7 L58. Haemolymph does not cover the wing surface. (...implying the outer surface.)

L38-39. It is incomplete and rather misleading to suggest the corrugations are responsible for aerodynamic performance when many other factors are also important. Even the role of corrugations is contentious (see discussions in Jongerius and Lentink (2010); Levy & Seifert (2009,2010) Bomphrey et al (2016); and others).

I am confused by the term "submicron" when referring to a film of 50um thickness?

It is not clear to me how the thin membranes were printed with a machine that has a resolution of 150um? This is likely thicker than the membrane. Nor is it immediately obvious how adding an extra layer will reduce stiffness (Page 5, L35).

P7, L46 C2. Consider adding recent paper by Salcedo & Socha (2020) to discussions/references relating to haemolymph.

Why do insect wings contain haemolymph? Does a bio-inspired wing necessarily require haemolymph? Why? Can we not simply engineer the stiffness to be the most appropriate stiffness? (Which, presumably for the insects is the un-desiccated stiffness.)

P8 L36. Replace “blood”.

The relative change in stiffness might be more informative than the absolute change. Also, ordering the wings by span, or body mass would be more useful to see if there is a trend with size.

Please provide detail of the mesh and “further refinement” of the FEA mesh in terms of cell number. It looks from the figure that there are three layers. Is this consistent throughout the model? What shape are the cells? Etc.

Please justify the constraint of the leading edge, which is curved, when testing chordwise flexibility.

Fig. 10. I’m not sure what to conclude from this figure/test. It would be useful to show the point of application of the load. It is not a particularly fair test of the effect of the insect wing membrane when the parts that have been removed are the same thickness as the veins. Membrane and veins have different behaviour in wings due to their shape and also their material properties. The effect of removing material is also difficult to assess given the change in range of the colour bar.

P11 L36. The claims of “superior performance” seem rather limited and premature. For example, the ability to deflect to great strains without damage is not superior if the wing is too compliant to support body weight and the real goal is to be stiff. This is addressed in the following paragraph in the next section, where the same wings are deemed to have failed.

The scale of the wings (3X larger than the model) should be made apparent in the methods.

Some acronyms are undefined on first use. (PIV; FDM)

Review form: Reviewer 2

Is the manuscript scientifically sound in its present form?

Yes

Are the interpretations and conclusions justified by the results?

Yes

Is the language acceptable?

Yes

Do you have any ethical concerns with this paper?

No

Have you any concerns about statistical analyses in this paper?

No

Recommendation?

Accept with minor revision (please list in comments)

Comments to the Author(s)

Overall, I believe the authors have done an exemplary job in preparing this manuscript. The level of scientific rigor is apparent, and the attention to detail regarding every aspect of the replication is appreciated.

I have a minor suggestion that the authors might consider, but it will not prevent moving forward.

1. Please add some quantitative results to the abstract.

Review form: Reviewer 3**Is the manuscript scientifically sound in its present form?**

Yes

Are the interpretations and conclusions justified by the results?

Yes

Is the language acceptable?

Yes

Do you have any ethical concerns with this paper?

No

Have you any concerns about statistical analyses in this paper?

No

Recommendation?

Accept with minor revision (please list in comments)

Comments to the Author(s)

The paper contains a high degree of novel content and is commendable for both breadth and depth. Analysis of these measurements form a structural engineering basis that is important for future flapping wing design and gliding flight characteristics. The test result demonstrates the feasibility of the wing prototypes mechanical properties and is a stepping-stone on the path to robotic locust-inspired gliding wing MAV.

At the beginning of this paper, the deficiencies in previous studies were clarified, and then the contribution of this study was proposed. Targeted and easy to compare. The author expresses the description of the content, very realistic.

However, there are still some deficiencies in the paper, which are recommended to modify.

1. There are some important recent and appropriate publications missing in this article as listed below:

-Reid, Heidi, et al. "Toward the design of dynamically similar artificial insect wings." *International Journal of Micro Air Vehicles* 13 (2021): 1756829321992138.

-Liu, Xiaohui, et al. "The Importance of Flapping Kinematic Parameters in the Facilitation of the Different Flight Modes of Dragonflies." *Journal of Bionic Engineering* 18.2 (2021): 419-427.

-Salami, Erfan, et al. "Nano-mechanical properties and structural of a 3D-printed biodegradable biomimetic micro air vehicle wing." IOP Conference Series: Materials Science and Engineering. Vol. 210. No. 1. IOP Publishing, 2017

-Timmermans, Siemen. "Modeling Flight Performance of Nano Air Vehicles with Flapping Wings." (2021).

2. Fabrication Methods is well presented in the manuscript and through the flowchart demonstration.

3. In section (e), the corrugation which has great importance and is clearly illustrated on the 3D printed wings for the readers. It is suggested to provide microscopic cut section view of the corrugation for better presentation to the readers.

4. To guide and inform the readers about the future of robotic insect flyers its strongly recommended to include recommendation for future works.

Decision letter (RSOS-202253.R0)

Dear Mr Isakhani

The Editors assigned to your paper RSOS-202253 "Towards locust-inspired gliding wing prototypes for micro aerial vehicle applications" have now received comments from reviewers and would like you to revise the paper in accordance with the reviewer comments and any comments from the Editors. Please note this decision does not guarantee eventual acceptance.

Please submit your revised manuscript and required files (see below) no later than 21 days from today's (ie 22-Apr-2021) date. Note: the ScholarOne system will 'lock' if submission of the revision is attempted 21 or more days after the deadline. If you do not think you will be able to meet this deadline please contact the editorial office immediately.

on behalf of Professor Brooke Flammang (Associate Editor) and R. Kerry Rowe (Subject Editor)
openscience@royalsociety.org

Reviewer comments to Author:

Reviewer: 1

Comments to the Author(s)

The authors present a study investigating the manufacturing feasibility and mechanical performance of bio-inspired insect wings. The model is a desert locust and the selection of wings are made by 3D printing, vacuum thermoforming, and laser cutting, or a combination of these methods.

The study refers to the authors' previous work using a Nash algorithm to optimise wing geometry beyond that seen in Nature given a limited set of objectives, although the link is not clear to the present work, the vein pattern is a reduced version of that found in the locust, scaled three fold. In a step towards testing the aerodynamic performance of models experimentally, and subsequently adding them to a glider, they explore several manufacturing techniques to identify the most appropriate against the criteria of accessibility, cost and structural performance. Aerodynamic tests and glider construction are not investigated here.

The manuscript describes a substantial amount of work. However, two sections seem rather out of place. There is a section on computational modelling which, while needing a little more methodological detail, seems to be reasonable. However, I did not see how it added to the story because the physical tests on prototypes are not used to validate the simulation, and they are also more comprehensive. There is also a section regarding haemolymph, but this does not seem to form part of the bio-inspired wing objective. These two sections disrupt the flow of testing wings made by different designs and manufacturing processes.

The manuscript concludes that one of the fabrication methods is better than the others but goes further to say it has "optimal" stiffness properties, which is not well supported.

I would invite the authors to consider what material properties would be ideally required for a manufactured glider (or flapping MAV). Specifically, what stiffness is optimal? This might well be different from the insect it mimics. Is flexion of the wing adaptive/beneficial in some way, or is the buckling simply a product of looking to minimise material/weight? If the latter, then would a stiffer wing be better? Or could a softer wing be ok too?

What is the material effect of haemolymph? Does it make a direct contribution to the bending properties (e.g. by pressurising the system) or is it an indirect effect, (e.g. by preventing desiccation of the cuticle)?

Because there are several fabrication methods, it is sometimes difficult to follow each through the workflow from the insect model start to engineered finish.

Specific comments:

"Superior structural performance of locusts." Measured against / superior to what?

L22. Veins do not 'coat' the wing surface.

P7 L58. Haemolymph does not cover the wing surface. (...implying the outer surface.)

L38-39. It is incomplete and rather misleading to suggest the corrugations are responsible for aerodynamic performance when many other factors are also important. Even the role of corrugations is contentious (see discussions in Jongerius and Lentink (2010); Levy & Seifert (2009,2010) Bomphrey et al (2016); and others).

I am confused by the term "submicron" when referring to a film of 50um thickness?

It is not clear to me how the thin membranes were printed with a machine that has a resolution of 150um? This is likely thicker than the membrane. Nor is it immediately obvious how adding an extra layer will reduce stiffness (Page 5, L35).

P7, L46 C2. Consider adding recent paper by Salcedo & Socha (2020) to discussions/references relating to haemolymph.

Why do insect wings contain haemolymph? Does a bio-inspired wing necessarily require haemolymph? Why? Can we not simply engineer the stiffness to be the most appropriate stiffness? (Which, presumably for the insects is the un-desiccated stiffness.)

P8 L36. Replace "blood".

The relative change in stiffness might be more informative than the absolute change. Also, ordering the wings by span, or body mass would be more useful to see if there is a trend with size.

Please provide detail of the mesh and "further refinement" of the FEA mesh in terms of cell number. It looks from the figure that there are three layers. Is this consistent throughout the model? What shape are the cells? Etc.

Please justify the constraint of the leading edge, which is curved, when testing chordwise flexibility.

Fig. 10. I'm not sure what to conclude from this figure/test. It would be useful to show the point of application of the load. It is not a particularly fair test of the effect of the insect wing membrane when the parts that have been removed are the same thickness as the veins. Membrane and veins have different behaviour in wings due to their shape and also their material properties. The effect of removing material is also difficult to assess given the change in range of the colour bar.

P11 L36. The claims of "superior performance" seem rather limited and premature. For example, the ability to deflect to great strains without damage is not superior if the wing is too compliant to support body weight and the real goal is to be stiff. This is addressed in the following paragraph in the next section, where the same wings are deemed to have failed.

The scale of the wings (3X larger than the model) should be made apparent in the methods.

Some acronyms are undefined on first use. (PIV; FDM)

Reviewer: 2

Comments to the Author(s)

Overall, I believe the authors have done an exemplary job in preparing this manuscript. The level of scientific rigor is apparent, and the attention to detail regarding every aspect of the replication is appreciated.

I have a minor suggestion that the authors might consider, but it will not prevent moving forward.

1. Please add some quantitative results to the abstract.

Reviewer: 3

Comments to the Author(s)

The paper contains a high degree of novel content and is commendable for both breadth and depth. Analysis of these measurements form a structural engineering basis that is important for future flapping wing design and gliding flight characteristics. The test result demonstrates the feasibility of the wing prototypes mechanical properties and is a stepping-stone on the path to robotic locust-inspired gliding wing MAV.

At the beginning of this paper, the deficiencies in previous studies were clarified, and then the contribution of this study was proposed. Targeted and easy to compare. The author expresses the description of the content, very realistic.

However, there are still some deficiencies in the paper, which are recommended to modify.

1. There are some important recent and appropriate publications missing in this article as listed below:

-Reid, Heidi, et al. "Toward the design of dynamically similar artificial insect wings."

International Journal of Micro Air Vehicles 13 (2021): 1756829321992138.

-Liu, Xiaohui, et al. "The Importance of Flapping Kinematic Parameters in the Facilitation of the Different Flight Modes of Dragonflies." Journal of Bionic Engineering 18.2 (2021): 419-427.

-Salami, Erfan, et al. "Nano-mechanical properties and structural of a 3D-printed biodegradable biomimetic micro air vehicle wing." IOP Conference Series: Materials Science and Engineering. Vol. 210. No. 1. IOP Publishing, 2017

-Timmermans, Siemen. "Modeling Flight Performance of Nano Air Vehicles with Flapping Wings." (2021).

2. Fabrication Methods is well presented in the manuscript and through the flowchart demonstration.

3. In section (e), the corrugation which has great importance and is clearly illustrated on the 3D printed wings for the readers. It is suggested to provide microscopic cut section view of the corrugation for better presentation to the readers.

4. To guide and inform the readers about the future of robotic insect flyers its strongly recommended to include recommendation for future works.

===PREPARING YOUR MANUSCRIPT===

===PREPARING YOUR REVISION IN SCHOLARONE===

<https://royalsociety.org/journals/authors/author-guidelines/#data>. You should ensure that

you cite the dataset in your reference list. If you have deposited data etc in the Dryad repository, please include both the 'For publication' link and 'For review' link at this stage.

Author's Response to Decision Letter for (RSOS-202253.R0)

See Appendix A.

Decision letter (RSOS-202253.R1)

Dear Mr Isakhani,

It is a pleasure to accept your manuscript entitled "Towards locust-inspired gliding wing prototypes for micro aerial vehicle applications" in its current form for publication in Royal Society Open Science. The comments of the reviewer(s) who reviewed your manuscript are included at the foot of this letter.

on behalf of Professor Brooke Flammang (Associate Editor) and R. Kerry Rowe (Subject Editor)
openscience@royalsociety.org

Appendix A

The authors would like to thank the Editor and Reviewers for all their time and effort spent on our first submitted manuscript. We are particularly grateful for all the compliments as well as the constructive comments that helped improve the manuscript in a number of ways such as;

- Precise referencing and citations.
- Amendments in figures and clarifications in result discussions.
- Normalising the calculated data to enhance readability of the results.
- Providing further clarification and description for the numerical analysis.
- And the overall coherence and consistency of the manuscript is greatly improved throughout with the help of the respected Referees' meticulous review comments.

➤ Response to Comments from the Reviewer-1

- 1) The authors present a study investigating the manufacturing feasibility and mechanical performance of bio-inspired insect wings. The model is a desert locust and the selection of wings are made by 3D printing, vacuum thermoforming, and laser cutting, or a combination of these methods. The study refers to the authors' previous work using a Nash algorithm to optimise wing geometry beyond that seen in Nature given a limited set of objectives, although the link is not clear to the present work, the vein pattern is a reduced version of that found in the locust, scaled three fold. In a step towards testing the aerodynamic performance of models experimentally, and subsequently adding them to a glider, they explore several manufacturing techniques to identify the most appropriate against the criteria of accessibility, cost and structural performance. Aerodynamic tests and glider construction are not investigated here.

Response: The Authors would like to thank the Reviewer for their interest in the topic and their careful consideration of the concepts presented in the article. And with regards to experimental aerodynamic analysis, you are absolutely right, we intend to complete this (PIV wind tunnel measurements) as a part of our future study.

- 2) The manuscript describes a substantial amount of work. However, two sections seem rather out of place. There is a section on computational modelling which, while needing a little more methodological detail, seems to be reasonable. However, I did not see how it added to the story because the physical tests on prototypes are not used to validate the simulation, and they are also more comprehensive. There is also a section regarding haemolymph, but this does not seem to form part of the bio-inspired wing objective. These two sections disrupt the flow of testing wings made by different designs and manufacturing processes.

Response: Thank you for this meticulous comment on the readability of our manuscript. As this is a very much valid suggestion, we made every effort to address the issue by adding clarifying statements in order to avoid such confusions in the future. Our additional statements included in the revised manuscript are; (1) page-8 para-4, that is a clarification on the fact that our Finite Element Analysis is a preliminary step to approve our digital designs prior to manufacturing, i.e. the designs with an FEA pass are considered worthy of manufacturing, so it is actually not really related to the physical tests although some relation was stated mistakenly in our first submission. (2) We believe our explanation hasn't been clear enough in relating the haemolymph effect to the rest of the study. Therefore, we have added a statement on page-6 para-5, to clarify the fact that the influence of haemolymph on wing flexibility plays a vital role in the process of bioinspired wing fabrication. In the sense, that if the natural wing's flexibility relies too much on the haemolymph flow, it would obviously be more difficult to fabricate such a wing since we are very far from being able to incorporate fluid flow inside our artificial wing designs. Therefore, if natural wings that we are being inspired by, gets its ideal flexibility-stiffness ratio solely from its structure, our chances of achieving that ratio would be higher. And our study on the haemolymph proves that the locust's wings are not the least dependent on haemolymph among other insects tested, so it is posed as a challenge to the researchers including us, focusing mainly on locust and dragonflies as a source of inspiration for the design of bioinspired artificial wings.

- 3) The manuscript concludes that one of the fabrication methods is better than the others but goes further to say it has "optimal" stiffness properties, which is not well supported.

Response: Thank you for spotting this ambiguous statement. We intended to claim that our final recommended fabrication method is capable of producing wing prototypes that maintain the best “trade-off strategy between flexibility and stiffness among the prototypes presented in the study” that also includes several examples from the literature. We strongly agree with the Reviewer that claiming “optimality” is not accurate and therefore we have removed this term throughout the manuscript. (highlighted in colour)

- 4) I would invite the authors to consider what material properties would be ideally required for a manufactured glider (or flapping MAV). Specifically, what stiffness is optimal? This might well be different from the insect it mimics. Is flexion of the wing adaptive/beneficial in some way, or is the buckling simply a product of looking to minimise material/weight? If the latter, then would a stiffer wing be better? Or could a softer wing be ok too?

Response: Thank you for your suggestions, these are indeed some of the trending questions related to our presented study. Of course the second question might well be related to the previous comment where we had mentioned the term “optimal stiffness”. We did agree that it is not an accurate description since stiffness is required at different levels for different applications. A balanced flexibility-stiffness ratio instead might be a more accurate description of our objective. That is, a fair balance of this ratio for an MAV wing might be the minimum requirement in order to be able to either glide or flap while also maintaining resilience in an event of a minor crash. We did try our best to include all the intriguing questions raised above by the respected Reviewer in our revised manuscript (in Section-3); some with potential answers/assumptions and some posed as a potential topic for a future study. (highlighted in colour)

- 5) What is the material effect of haemolymph? Does it make a direct contribution to the bending properties (e.g. by pressurising the system) or is it an indirect effect, (e.g. by preventing desiccation of the cuticle)?

Response: Thank you for your comment, this topic is indeed very intuitive. Actually there is very limited literature on this subject unfortunately. Although we do intend to carry forward this study using SEM imaging and mechanical testing to better understand the underlying physiological properties related to the haemolymph flow in various insect wings. However, at our current stage of observation we believe both of your assumptions are valid since the fluid flow can be responsible for both pressurising as well as moisturising the wing cuticles. This might well be a matter of concern for many of our readers too, therefore we added a statement including several hints for a potential future study.

- 6) Because there are several fabrication methods, it is sometimes difficult to follow each through the workflow from the insect model start to engineered finish.

Response: That is absolutely correct. The flow is a bit confusing since there are interconnections between different processes at times which is inevitable due to their hybrid composition. We have tried to present an overall picture of the flow in Figure-1 to distinguish between each method. This figure is improved significantly to increase the readability of the revised article.

- 7) “Superior structural performance of locusts.” Measured against / superior to what?

Response: Thank you for spotting another vague superlative. We have clarified this by adding specific comparative subjects. (highlighted in colour)

8) L22. Veins do not ‘coat’ the wing surface.

Response: That is absolutely correct, we apologise for the language error. The words that we intended to use were “spread over”. Therefore, the statement is amended accordingly. (highlighted in colour)

9) P7 L58. Haemolymph does not cover the wing surface. (...implying the outer surface.)

Response: Apologies again, “cover” is another inaccurate term used in this sentence. We have amended this as “”. (highlighted in colour)

10) L38-39. It is incomplete and rather misleading to suggest the corrugations are responsible for aerodynamic performance when many other factors are also important. Even the role of corrugations is contentious (see discussions in Jongerius and Lentink (2010); Levy & Seifert (2009,2010) Bomphrey et al (2016); and others).

Response: Yes, that is absolutely correct. We strongly agree with the Reviewer on this issue. There should be an emphasis on the fact that corrugations might be ONE of the responsible factors and not the sole contributor of the locust’s aerodynamic performance. This statement is amended according to your helpful suggestion.

11) I am confused by the term “submicron” when referring to a film of 50um thickness?

Response: Thank you for your meticulous observation. That is indeed an erroneous representation of the number since submicron means less than 1µm. This is now amended in the revised manuscript.

12) It is not clear to me how the thin membranes were printed with a machine that has a resolution of 150um? This is likely thicker than the membrane. Nor is it immediately obvious how adding an extra layer will reduce stiffness (Page 5, L35).

Response: You are absolutely correct. These sentences are slightly confusing and needs amendments. We have done this by clearly stating that the 150µm PVC sheets are not printed, they are basically off the shelf thin PVC sheets available for automobile modification purposes commonly called vinyl wraps. Only the wing venations are 3D printed which are 650µm as also shown in Figure-3. Also, the error in the statement (in L35) is correctly pointed out by the Reviewer. Addition of the membrane only increases resilience of the wing and not its flexibility according to our mechanical tests. These are amended in the revised manuscript accordingly.

13) P7, L46 C2. Consider adding recent paper by Salcedo & Socha (2020) to discussions/references relating to haemolymph. Why do insect wings contain haemolymph? Does a bio-inspired wing necessarily require haemolymph? Why? Can we not simply engineer the stiffness to be the most appropriate stiffness? (Which, presumably for the insects is the undesiccated stiffness.)

Response: Thank you for your suggestions. We have cited the aforementioned references in the revised manuscript. The questions regarding the haemolymph study are truly intriguing and needs more attention. Although we intend to continue this study in another dedicated article, we raised these challenging questions (as a potential topic for future study) in our revised manuscript following the respected Reviewer's suggestion.

- 14) P8 L36. Replace "blood". The relative change in stiffness might be more informative than the absolute change. Also, ordering the wings by span, or body mass would be more useful to see if there is a trend with size.

Response: Thank you for the precise identification of this inappropriate terminology. This is now amended in the revised manuscript. EI for different insects is now represented in percentage change which shed an entirely new light on our observations and conclusions made in the haemolymph study. Order of the wings are also changed and sorted according to wingspan following your clear instructions.

- 15) Please provide detail of the mesh and "further refinement" of the FEA mesh in terms of cell number. It looks from the figure that there are three layers. Is this consistent throughout the model? What shape are the cells? Etc. Please justify the constraint of the leading edge, which is curved, when testing chordwise flexibility.

Response: We strongly agree with the Reviewer on the fact that our FEA section lacked sufficient explanation. We have tried our best to address all the issues raised and add further description to the analysis including amendments made in figure-9. (highlighted in colour)

- 16) Fig. 10. I'm not sure what to conclude from this figure/test. It would be useful to show the point of application of the load. It is not a particularly fair test of the effect of the insect wing membrane when the parts that have been removed are the same thickness as the veins. Membrane and veins have different behaviour in wings due to their shape and also their material properties. The effect of removing material is also difficult to assess given the change in range of the colour bar.

Response: Thank you for your useful suggestions. We have updated the figure-10 with your suggested amendments. However, some of the issues raised are unfortunately limitations of the algorithm used. For instance, the thickness of the wing venations and membrane being same is resulted from the fact that for a digital wing, thickness of the venations is the least thickness that can be achieved while also maintaining corrugations. In other words, the submicron-thick membranes of the insect wing can't be digitally modelled while being corrugated, it can only be modelled if it is considered as a flat plate. This can be checked on the CAD models attached as supplementary files. And since corrugations were more important to our study, we didn't want to eliminate that feature. Furthermore, this is actually practiced in several related literatures as well. And with regards to the information laid in Figure-10, unfortunately the scale is limited to the solutions obtained. In other words, the maximum value (that the user defines) cannot be lower than maximum value from the results obtained by the FEA solution and same holds good for minimum values therefore the resulting figures don't really change after editing legend scale. And since ANSYS is a commercial software this is an embedded limitation. However, we can confirm that the comparative figure-10 could address two issues; (1) that the removal of such a thick membrane (as you have mentioned) does not really affect the mechanical performance of the digitised model by a large margin, and (2) the fact that the venation pattern would be stable enough for fabrication i.e., manufacturing-worthiness of the digitised wing is established. These are the clarifying statements added in FEA section of the revised manuscript.

17) P11 L36. The claims of “superior performance” seem rather limited and premature. For example, the ability to deflect to great strains without damage is not superior if the wing is too compliant to support body weight and the real goal is to be stiff. This is addressed in the following paragraph in the next section, where the same wings are deemed to have failed.

Response: Thank you for your precise observation, this is another inappropriate usage of a superlative. We have amended this in the revision according to your instructions.

18) The scale of the wings (3X larger than the model) should be made apparent in the methods.

Response: Yes, that is absolutely correct. We have reiterated this in the material and methods section of the revised manuscript.

19) Some acronyms are undefined on first use. (PIV; FDM)

Response: Thank you for spotting the readability issue. We have proofread the article to identify and amend all such undefined acronyms, typos, and compilation errors. (highlighted in colour)

Response to Comments from the Reviewer-2

1) Overall, I believe the authors have done an exemplary job in preparing this manuscript. The level of scientific rigor is apparent, and the attention to detail regarding every aspect of the replication is appreciated. I have a minor suggestion that the authors might consider, but it will not prevent moving forward.

a) Please add some quantitative results to the abstract.

Response: Thank you very much for your kind compliments. We deeply appreciate your time and efforts spent in carefully considering our manuscript. Your precise suggestion is implemented in this revision and the article is amended according to your helpful instruction. (highlighted in colour)

Response to Comments from the Reviewer-3

- 1) The paper contains a high degree of novel content and is commendable for both breadth and depth. Analysis of these measurements form a structural engineering basis that is important for future flapping wing design and gliding flight characteristics. The test result demonstrates the feasibility of the wing prototypes mechanical properties and is a stepping-stone on the path to robotic locust-inspired gliding wing MAV. At the beginning of this paper, the deficiencies in previous studies were clarified, and then the contribution of this study was proposed. Targeted and easy to compare. The author expresses the description of the content, very realistic. However, there are still some deficiencies in the paper, which are recommended to modify.

Response: The Authors would like to thank you for your careful consideration of our manuscript and your kind compliments. Your constructive comments are strictly implemented and the amendments made in this revision accordingly.

- 2) There are some important recent and appropriate publications missing in this article as listed below:
-Reid, Heidi, et al. "Toward the design of dynamically similar artificial insect wings." International Journal of Micro Air Vehicles 13 (2021): 1756829321992138.
-Liu, Xiaohui, et al. "The Importance of Flapping Kinematic Parameters in the Facilitation of the Different Flight Modes of Dragonflies." Journal of Bionic Engineering 18.2 (2021): 419-427.
-Salami, Erfan, et al. "Nano-mechanical properties and structural of a 3D-printed biodegradable biomimetic micro air vehicle wing." IOP Conference Series: Materials Science and Engineering, Vol. 210. No. 1. IOP Publishing, 2017
-Timmermans, Siemen. "Modeling Flight Performance of Nano Air Vehicles with Flapping Wings." (2021).

Response: Thank you very much for your highly useful suggestion. We have carefully cited the above mentioned literature in the revised manuscript.

- 3) Fabrication Methods is well presented in the manuscript and through the flowchart demonstration.

Response: The Authors truly appreciate your time and efforts spent on our presented article. We hope our revision could further satisfy the standard requirements of the RSOS.

- 4) In section (e), the corrugation which has great importance and is clearly illustrated on the 3D printed wings for the readers. It is suggested to provide microscopic cut section view of the corrugation for better presentation to the readers.

Response: This is indeed a perfect suggestion, as the microscopic view of the wings' cross-section profiles is highly intuitive and informative for the readers. For this we have provided the Figure-2 which is actually a hybrid figure (i.e. a combination of illustrations + microscopic images + digitised wing model) representing the different graphics merged onto one single figure. We have now added clarifying statements in the revision to point this out and hopefully avoid any future ambiguity.

- 5) To guide and inform the readers about the future of robotic insect flyers its strongly recommended to include recommendation for future works.

Response: The Authors strongly agree with the Reviewer that suggestions for potential future work is a vital element of a credible research article. Therefore, we have highlighted several suggestions in the conclusion section of the revised manuscript in cyan coloured text.

The Authors would like to take this opportunity to appreciate the Reviewers' time and efforts spent in providing the valuable and constructive comments which helped polish and add consistency to the submitted manuscript significantly.